# The Value of Clinical Decision Support in Healthcare: A Focus on Screening and Early Detection

**DOI:** 10.3390/diagnostics15050648

**Published:** 2025-03-06

**Authors:** Hendrik Schäfer, Nesrine Lajmi, Paolo Valente, Alessandro Pedrioli, Daniel Cigoianu, Bernhard Hoehne, Michaela Schenk, Chaohui Guo, Ruby Singhrao, Deniz Gmuer, Rezwan Ahmed, Maximilian Silchmüller, Okan Ekinci

**Affiliations:** 1Clinical Development & Medical Affairs, Roche Diagnostics International Ltd., Forrenstrasse 2, 6343 Rotkreuz, Switzerlandruby.singhrao@roche.com (R.S.); 2Medical Faculty, Friedrich Schiller University Jena, 07737 Jena, Germany; 3Clinical Value & Validation, Roche Information Solutions, 2881 Scott Blvd, Santa Clara, CA 95050, USA; 4Clinical Value & Validation, Roche Information Solutions, F. Hoffmann-La Roche Ltd., Grenzacherstrasse 124, 4070 Basel, Switzerland; 5Quality & Regulatory Roche Information Solutions, Roche Diagnostics International Ltd., Forrenstrasse 2, 6343 Rotkreuz, Switzerland; 6Healthcare Insights, Roche Information Solutions, Roche Diagnostics International Ltd., Forrenstrasse 2, 6343 Rotkreuz, Switzerland; 7Data, Analytics & Research, Roche Information Solutions, 2881 Scott Blvd, Santa Clara, CA 95050, USA; 8Wiener Gesundheitsverbund, Klinik Landstraße, Juchgasse 25, 1030 Vienna, Austria; 9Digital Technology & Health Information, Roche Information Solutions, 2841 Scott Blvd, Santa Clara, CA 95050, USA; 10School of Medicine, University College Dublin, D04 C1P1 Dublin, Ireland

**Keywords:** clinical decision support, algorithms, critical care, cardiology, nephrology, oncology

## Abstract

In a rapidly changing technology landscape, “Clinical Decision Support” (CDS) has become an important tool to improve patient management. CDS systems offer medical professionals new insights to improve diagnostic accuracy, therapy planning, and personalized treatment. In addition, CDS systems provide cost-effective options to augment conventional screening for secondary prevention. This review aims to (i) describe the purpose and mechanisms of CDS systems, (ii) discuss different entities of algorithms, (iii) highlight quality features, and (iv) discuss challenges and limitations of CDS in clinical practice. Furthermore, we (v) describe contemporary algorithms in oncology, acute care, cardiology, and nephrology. In particular, we consolidate research on algorithms across diseases that imply a significant disease and economic burden, such as lung cancer, colorectal cancer, hepatocellular cancer, coronary artery disease, traumatic brain injury, sepsis, and chronic kidney disease.

## 1. Introduction

Clinical Decision Support (CDS) systems have become essential in modern healthcare, increasing patient management through Artificial Intelligence (AI) and machine learning (ML). By processing complex data, these systems provide real-time support in diagnosis and personalized treatment while addressing human limitations in multi-dimensional care decisions. In the first part of this literature review, we provide a comprehensive analysis of CDS systems, emphasizing their role in early detection and prevention. In the second part, we discuss selected diseases with a high disease burden whose prevention, by using AI, may substantially decrease mortality, prevalence, Years Lived with Disability (YLD), and healthcare costs. We display various commercially and non-commercially available algorithms in the context of their performance parameters and how they compare to the generally accepted standard of care.

### 1.1. Artificial Intelligence, Machine Learning and Clinical Decision Support

Because human cognition is inherently limited when processing multifactorial factors, causal connections can be challenging for the human mind to grasp [1]. AI is a branch of computer science that enables machines to perform tasks based on large amounts of data. AI can quickly process test variables that may initially seem irrelevant but could reveal significant and complex health-related insights [2]. In addition, AI efficiently processes and analyzes data in real time, thereby saving time and resources [3]. A subset of the AI umbrella term is ML, which enables machines to harvest knowledge and learn independently using the acquired data [4]. There are three main types of ML: Supervised learning: This involves labeled training data to learn and make predictions. Examples include support vector machines, decision trees, random forests, and logistic regression;Unsupervised learning: This involves working with unlabeled data to identify patterns and relationships. Common techniques include clustering algorithms like K-Means and Gaussian Mixture Models;Reinforcement learning: This involves learning through interactions with an environment and receiving rewards or penalties to optimize decision-making. Applications include robotics, game-playing, and autonomous driving.

A combination of unsupervised and supervised learning is used by generative AI (GenAI). Also referred to as semi-supervised learning, new content in text, images, or audio can be created. This is based on patterns learned from labeled and unlabeled training data. In 2023, Google announced its collaboration with Mayo Clinic on the “Enterprise Search on Generative AI App Builder”, a tool allowing physicians to create chatbots [5]. The tool can extract data from electronic health records that are stored in multiple locations and create reports for individual use cases. Whilst GenAI may contribute to operational effectiveness in administration, concerns arise around content accuracy and the potential risk for patients [6].

### 1.2. Artificial Intelligence as Clinical Decision Support: Why Is It Needed?

The complexity of medicine itself, high patient turnover, or shift changes [7] inter alia are root causes [8] of malpractice. Incorrect application of drugs, for example, occurs in 6.5 out of 100 cases and is the most common but preventable cause of patient injury [9]. Annually, medication errors lead to 7000 and 9000 deaths in the US alone [10]. Of note, a cross-sectional analysis indicated that only 15 diseases account for approximately 50.7% of all severe treatment mistakes [11]. The study revealed that five diseases—stroke, sepsis, pneumonia, venous thromboembolism, and lung cancer—contribute to 39% of malpractice costs. Because the complexity of medical conditions is positively correlated to poor outcomes [12], AI can be employed to improve decisions when multimodal data (EMR (Electronic Medical Record) and LIS (Laboratory Information System)) are required [13]. CDS systems compare patient data with previously collected data sets of similar casuistics to calculate a desired endpoint [14]. The success of CDS is primarily based on the magnitude of training data and validation cohorts.

### 1.3. Applications of CDS in Practice

CDS provides benefits in the following settings:Improvement of administration: CDS systems can support clinicians in coding [15]. For example, using an anatomical interface representing the human body in an emergency department (ED) led to a faster selection of admission codes [16]. Another study yielded an improvement in the documentation accuracy of “induction of labor” when a prompting system was offered to the physician [17];Adherence to evidence-based medicine: CDS can improve adherence to medical guidelines [18], thus preventing inadequate treatment and helping to maintain monitoring intervals for check-ups when patient cooperation is limited [19];Improvement of patient safety: CDS effectively reduces prescription errors [20] and supports the right choice of antibiotics [21]. Furthermore, use cases exist for a CDS-based avoidance of nephrotoxic substances [22]. The University of Heidelberg, for example, introduced a CDS to translate drug names into the names of the clinic’s pharmacy because one of five substitutions was incorrect [23]. CDS can also trigger alarms in a resource-constrained monitoring environment [24]. In 2018, the University of Leipzig implemented an automated warning system called AMPEL (Analyse- und Meldesystem zur Verbesserung der Patientensicherheit durch Echtzeitintegration von Laborbefunden) that uses more than 10 validated algorithms and has led to more than 6000 alarms per year. Originally designed to create alarms for Refeeding syndrome, AMPEL has developed into an open-source project from 2024 onward;Physician-facing CDS: AI-based software can generate possible diagnoses and their related differential diagnosis [25]. In a randomized controlled trial with 87 general practitioners, CDS-assisted diagnoses were generated with 82% accuracy. This translates into an almost 10% better diagnostic accuracy rate as compared to medical colleagues who did not use the software [26]. Improvements in diagnostic accuracy are evident for other CDS systems, too, such as those from Smart Blood Analytics [27]. One of their algorithms classifies hematological conditions. Although the clinical judgment of hematologists and internal specialists was still slightly better than the performance of the algorithm, non-hematology internal medicine specialists achieved an accuracy of 0.26 compared to an accuracy of 0.60 provided by the CDS [28]. Physician-facing CDS can be applied in various stages of the diagnostic workup. Table 1 displays examples of academically or commercially available algorithms in different stages of the patient journey, which correspond to the three levels of prevention defined by Leavell and Clark [29];Patient-facing CDS: Empowering patients to control aspects of their care is particularly important, as it improves adherence to recommendations [30]. Ada Health, for example, is an AI-powered symptom assessment tool for patients used by over 12 million users. The tool generates possible causes of symptoms and recommends next steps [31].

**Table 1 diagnostics-15-00648-t001:** Overview of selected algorithms according to setting, prevention level, and clinical use.

Stage of Patient Journey	Prevention Level Based on Leavell and Clark	Intention	Examples: Companies and Algorithms	Intended Use
Risk Stratification for Screening [32]	Secondary Prevention	Prognostic enrichment of patient population eligible for screening with the intention of prioritization	Medial EarlySign (ColonFlag™)	Identification of patients at high risk for colorectal cancer by analyzing age, sex, and a recent complete blood count
Screening [33]	Secondary Prevention	Algorithms determine whether a particular disease is present or the patient is at risk of developing it	GUARDANT (Shield)	Qualitative, in vitro diagnostic test to detect colorectal cancer-derived alterations in cell-free DNA from blood, indicated in individuals at average risk for age 45 years or older
Diagnosis [34]	Secondary Prevention	Algorithm determining the underlying root cause of disease	Saventic Health (SARAH)	Diagnosis of rare diseases using natural language processing and AI/ML
Classification [35,36]	Secondary Prevention	Algorithms to classify diseases of similar phenotypes and/or staging	Deep6A	Matching patients and sites to actual trial protocols in real time
			Smart Blood Analytics (Virus vs. Bacteria)	Differentiation between viral and bacterial infections based on 17 blood tests, sex, and age
Prognosis [37]	Secondary Prevention	Algorithms determining how likely a particular outcome will be reached	AlgoDx (NAVOY Sepsis)	Forecasting patients’ risk of sepsis 3 h before traditional rule-based scoring systems
Response Prediction [38]	Secondary Prevention	Algorithms predicting the likelihood of a drug to have a therapeutic effect on an individual	SPHINKS	Probability assessment of therapeutic response with glioblastoma kinases and to inform patient selection in prospective clinical trials
Monitoring [39]	Secondary Prevention	Algorithms assessing relapse or worsening of preexisting conditions.	Lenus (Stratify)	AI-based risk scores for hospital admission and readmission. Risk scores provide care teams with actionable insights to help inform appropriate treatment interventions, keeping patients at home and out of the hospital

Physician-facing CDS can be applied in various stages of the diagnostic workup. Table 1 displays examples of academically or commercially available algorithms in different stages of the patient journey, which correspond to the three levels of prevention defined by Leavell and Clark [33].

### 1.4. Quality Metrics of CDS

User acceptance of CDS, regardless of whether the user is a physician, a nurse, or a patient, is a fundamental requirement for its successful application [40]. Achieving this requires evaluating CDS systems against comprehensive quality metrics that ensure technical robustness, clinical relevance, and ethical integrity. A critical aspect of this evaluation involves identifying specific clinical challenges and designing performance metrics that enhance the utility of these systems for clinicians [41]. This process necessitates close collaboration with healthcare professionals to optimize CDS for real-world application. Addressing diverse objectives and priorities involves implementing multiple functions and constraints—such as diagnostic accuracy, timeliness, healthcare costs, and resource capacity.

Furthermore, these metrics are vital in identifying and mitigating biases, ensuring equitable performance across patient populations. The following table displays seven features considered a prerequisite for ideal algorithms applied in any form of CDS [42] (Table 2). These features and metrics serve as a foundation for developing, evaluating, and refining CDS systems.

### 1.5. Challenges and Limitations of CDS

Despite its advantages, CDS in healthcare faces limitations and legal concerns [46]. Addressing them effectively ensures successful implementation in clinical practice. We consider the following five aspects as relevant:Data protection, security, and compliance: Data protection involves stringent security measures to safeguard patient information and adhere to legal standards, such as the US Health Insurance Portability and Accountability Act Part 160, 162, and 164 (Administration) or the European General Data Protection Regulation [47]. Compliance includes the use of advanced encryption technologies and secure servers. Medical device companies must comply with guidelines set by regulatory authorities such as the FDA [48] and the EU Medical Device Regulation [49]. Other frameworks, such as the EU AI Act [50], the world’s first comprehensive AI law, ensure that risk is mitigated and ethical standards are met;Usability: The acceptance of CDS depends heavily on its usability. User-centered design facilitates the integration of algorithms into daily practice with minimal disruption to existing workflows [1]. Early collaboration with future CDS users can help improve user interfaces. Training enables physicians to use the technology effectively, make informed decisions, and understand its limitations [51];Clinical validation: Access to independent cohorts after an algorithm’s development is often limited. To increase the acceptance of CDS, models must be validated across ethnicities and age groups. Real-world evidence can enhance trust and demonstrate robustness in clinical environments;Information overload: The abundance of data provided by CDS software can be overwhelming for medical professionals and may lead to “information fatigue”. Effective data visualization can help manage the flood of information [52]. CDS should only present patient data relevant to the respective context;Generative AI models, which are increasingly integrated into CDS, are prone to hallucinations—instances where the AI generates incorrect or fabricated information. Contemporary research suggests that hallucinations should be labeled as “confabulations” or, better yet, as AI misinformation to prevent the stigmatization of AI [53]. In a clinical context, misinformation could result in false interpretation of patient data or the suggestion of invalid medical diagnoses and treatments [54]. Addressing AI-based misinformation requires robust verification mechanisms, cross-referencing AI outputs with established medical knowledge, and ensuring that human oversight remains central to decision-making. Also, a model with more parameters trained for longer tends to confabulate less. However, this is computationally expensive and involves trade-offs with other chatbot skills, such as the ability to generalize. Training on larger, cleaner data sets helps, but there are limits to what data are available. Developers must prioritize transparency in how generative models operate and train them with high-quality, domain-specific datasets to minimize errors. Furthermore, regulatory frameworks should mandate rigorous testing and validation of generative AI components in CDS systems.

#### Summary: CDS in Clinical Practice

CDS utilizes various ML approaches—supervised, unsupervised, and reinforcement learning—to learn from data, identify patterns, make predictions, and optimize decision-making. They assist clinicians, e.g., by streamlining administrative tasks, providing diagnostic support, ensuring adherence to evidence-based guidelines, reducing prescription errors, and enhancing patient safety with timely alerts and monitoring.

The successful implementation of CDS depends on user acceptance and quality. Ideal algorithms are explainable, accurate, adaptable to new data, autonomous yet allow for human oversight, fair across diverse populations, and reproducible in different settings. Medical value is an additional feature developers should consider to ensure commercialization and desirability. Challenges include ensuring data security and compliance with regulations like HIPAA and GDPR, designing user-friendly interfaces that integrate seamlessly into clinical workflows, and overcoming resistance to change by emphasizing CDS as supportive tools that augment clinical judgment.

Regular monitoring will ensure algorithms remain accurate without confabulation as clinical practices and patient demographics evolve. Close collaboration among clinicians, technologists, and policymakers in this field is vital to fully and sustainably realizing these systems’ potential to improve outcomes and shape the future of healthcare.

## 2. Algorithm Types of CDS Through the Lens of Prevention

While algorithms on the primary, secondary, and tertiary prevention levels [55] are desirable, their development complexity varies significantly. Algorithms for primary prevention are challenging in design since they focus on a pre-disease state [56]. Algorithms in secondary prevention target pre-symptomatic diagnosis of diseases with either curative intent or a delay in disease progression, which ultimately leads to a higher likelihood of cure and subsequently lower healthcare costs. Classical applications are screening and early detection. Algorithms in tertiary prevention are applied when a disease is evident or treated to affect the condition positively. In this section, we present contemporary algorithms for secondary prevention of eight diseases, such as lung cancer, colorectal cancer (CRC), hepatocellular cancer (HCC), liver fibrosis, coronary artery disease (CAD), traumatic brain injury (TBI), sepsis, and chronic kidney disease (CKD). Those conditions were selected based on the global burden of disease because of their significance in prevalence, mortality, YLD, and healthcare costs.

Ample evidence exists that secondary prevention can reduce mortality of malignancies by 15 to 20% [57]. Because lung cancer (18.7% of all cancer deaths), CRC (9.3%), and liver cancer (7.8%) remain the leading causes of cancer mortality globally [58], there is an unmet medical need to improve early detection. A 2017 US study found that early detection can lead to cost savings of USD 3.47B for CRC and USD 3.43B for lung cancer [59].

On a global scale, CAD remains the leading cause of death, accounting for approximately 13% of the world’s total deaths [60]. In a global comparison of economic costs, annual expenses for the condition exceeded the total health expenditure per capita by 4.9% to 137.8% [61], thus qualifying for early detection and consecutive intervention. 

Liver fibrosis algorithms, which integrate biomarkers and clinical data, offer a promising alternative to liver biopsies for assessing liver disease progression. Some of these algorithms have received FDA approval and CE marking, indicating that they meet certain regulatory standards for clinical use. However, their effectiveness can still be variable, as performance often depends on factors such as the underlying liver disease, patient demographics, and the specific biomarkers incorporated.

TBI implies approximately USD 400 billion in annual costs to healthcare [62] and will remain the leading cause of disability beyond 2030. Despite implying a higher healthcare burden as compared to neurodegenerative disorders, the condition goes almost unrecognized in public health strategies [63].

Sepsis implies approximately 20% of all global deaths [64], of which 50% occur in children between 1 and 4 years [65]. Because the condition is considered the leading cause of death in critically ill patients, with an almost 50% mortality [64], the implication on healthcare spending is significant and contributes to 2.65% of the entire healthcare budget [66]. As a result, the WHO recognized sepsis as a major threat to global health [64,67] and highlighted the necessity for improved prevention.

Finally, we included chronic kidney disease (CKD) as a relevant condition in our analysis because approximately 850 million patients worldwide suffer from it, most of them in low- and lower-middle-income countries with no access to diagnosis and treatment [68]. Since CKD is the third fastest-growing cause of death globally [69], the condition and its epidemiology qualify well for early detection by innovative algorithms, especially given the increasing availability of point-of-care devices.

Table 3 provides an overview of contemporary algorithms for diseases in focus, describing information on the algorithm’s purpose, the parameters used, the application, sensitivity, specificity, area under the curve (AUC), limitations, regulatory status, and clinical use.

### 2.1. Oncology

Cancer is among the leading causes of death globally, with 9.7 million deaths, according to recent estimates from the World Health Organization’s International Agency for Research on Cancer. This number is anticipated to double, reaching 18.5 million by 2050. With ~50% of the global number of new cases, Asia leads a significant portion of this burden. The most common cancers in Asia are lung, breast, and CRC, while lung, liver, and gastric cancers were the leading causes of death, as reported in 2022 [70]. East Asia and the Pacific bear the highest aggregate economic cost of cancer, totaling USD 9.7 trillion [71]. Screening large numbers of asymptomatic individuals through an organized screening program effectively reduces mortality, morbidity [72], and incidence [73]. Numerous Asian countries have integrated cancer screening programs into their national cancer control plans [74].

Nevertheless, screening initiatives fail to deliver consistent benefits across different socioeconomic and demographic groups [75], mainly due to the complexity of real-world screening as it involves multiple steps. The initial step identifies asymptomatic individuals eligible for screening, the initial screening test, timely diagnosis, and treatment. However, breakdowns occur at any point in this continuum, reducing the overall effectiveness and potential of the screening program.

**Table 3 diagnostics-15-00648-t003:** Contemporary algorithms for secondary prevention of eight relevant disease areas: (a): lung cancer; (b): colorectal cancer; (c): hepatocellular cancer diagnostics; (d): liver fibrosis; (e): cardiology; (f): traumatic brain injury; (g): sepsis; and (h): chronic kidney disease. The algorithms displayed are not exhaustive.

	Algorithm Name	Purpose	Method/Parameters Used	Application	Sensitivity	Specificity	AUC	Limitations	Regulatory Status	Clinical Use
a	Bach Model [76,77]	To estimate the 10-year absolute risk of lung cancer among individuals based on their clinical history and exposure factors.	Cox proportional hazard regression to estimate multivariable relations. Predictors include age, sex, smoking history/abstinence, family history of lung cancer, secondary smoke exposure, and asbestos exposure.	Assessing variability in lung cancer risk within high-risk populations.	84.0%	65.0%	0.739–0.824	Only tested in high-risk CS/FS; Complicated data collection for asbestos exposure; may not be suitable for lung cancer screening programs.	Not yet FDA-approved or CE-marked.	Supports decision-making for lung cancer screening and helps stratify participants for clinical trials.
a	Spitz Mode [76,78]	To predict the 1-year probability of developing lung cancer in never, former, and current smokers.	Multivariable logistic regression models stratified by smoking status (never, former, and current smokers). Variables included a history of hay fever, family history of lung cancer, secondary smoke exposure, asbestos exposure, dust exposure, pneumonia (previous diagnosis), and COPD.	Identification of individuals at high risk for lung cancer who might benefit from increased surveillance or preventive interventions. Support for the design of clinical trials targeting high-risk populations.	Reported as true-positive rates: - Never smokers: Not explicitly provided; - Former smokers: Approximately 70%; - Current smokers: Approximately 68–69%	Reported as true-negative rates: -Former smokers: Approximately 66%; -Current smokers: Approximately 65–68%	Never smokers: 0.47–0.66 Former smokers: 0.58–0.69 Current smokers: 0.52–0.64	Derived from a single case–control study at a specific cancer center; it may not generalize to broader populations focused only on non-Hispanic white participants. Limited discriminatory power (modest AUC values).	Not specified as an FDA-approved or regulated tool; primarily for research and clinical trial designs.	Helps in counseling high-risk individuals for screening or preventive measures. Aids in clinical trial enrollment by identifying participants with high predicted lung cancer risk.
a	Liverpool Lung Project Model [76,79]	To estimate an individual’s 5-year absolute risk of developing lung cancer based on a combination of risk factors.	Multivariate logistic regression model using conditional logistic regression. Risk factors included age, gender, smoking duration, family history of lung cancer (early and late onset), occupational exposure to asbestos, prior diagnosis of malignant tumors, and prior diagnosis of pneumonia. The absolute risk was calculated by combining the logistic model with regional lung cancer incidence data.	Identification of high-risk individuals for lung cancer screening. Guidance for primary care clinicians in patient risk assessment. Potential integration into early detection strategies like CT screening.	62% at a 2.5% cutoff for identifying high-risk individuals 34% at a 6% cutoff	70% at a 2.5% cutoff 90% at a 6% cutoff	0.698–0.790	Based on case–control data from a single geographic region, which may limit generalizability. Evidence for accurate prediction in people who do not smoke is lacking. Recall bias due to reliance on self-reported data for risk factors like smoking and asbestos exposure.	Not specified as a regulated tool; primarily a research and risk stratification model.	Assists in risk stratification for lung cancer screening and prevention strategies. Provides an evidence-based approach for identifying high-risk individuals who may benefit from targeted interventions.
a	PLCOm2012 Model [72,80]	To predict lung cancer risk over 6 years to optimize the selection of individuals for lung cancer screening, improving sensitivity and efficiency compared to traditional categorical criteria like USPSTF2013.	A multivariable logistic regression model incorporating age, smoking history (intensity, duration, quit years), race/ethnicity, BMI, family history of lung cancer, personal history of cancer, history of COPD, and education level.	Stratifying individuals for lung cancer screening to maximize early detection and cost-effectiveness. Clinical and public health programs to identify high-risk populations.	85.3%	65.6%	0.699–0.803	Validation is limited to specific cohorts; external generalizability may require further studies. Requires collection of detailed patient data, which could complicate implementation,	Not a regulated tool but is recommended for use in lung cancer screening programs in some countries (e.g., the UK and Canada).	Incorporated in screening programs in Canada and the UK and proposed for others. Guides clinical decisions by identifying high-risk individuals beyond standard age-pack-year criteria.
a	LungFlag™ [81]	To identify patients at risk of developing lung cancer up to 12 months before clinical diagnosis.	Machine learning developed approach based on Extreme Gradient Boosting (XGBoost). Data sources included sociodemographic factors, smoking history, laboratory results (e.g., complete blood count), and clinical history (e.g., history of COPD, BMI, etc.).	Screening high-risk individuals for early detection of lung cancer.	40.1%	95%	0.841–0.871	The model is most accurate closer to the time of diagnosis (0–3 months), which may reflect the onset of clinical suspicion rather than purely pre-symptomatic detection. Missing or incomplete smoking data impacted model performance.	Not yet CE marked, FDA- exempt	Proposed for identifying high-risk individuals for early lung cancer detection.
b	Yang Model [82]	Predicts the risk of ACN in asymptomatic adults, including younger populations (<50 years) often excluded from routine screening.	Developed using logistic regression on clinical and laboratory parameters such as age, sex, family history, body mass index, smoking, serum fasting glucose, LDL, and carcinoembryonic antigens.	Risk stratification for ACN. Guides selection of CRC screening methods: - High risk: Colonoscopy; - Borderline risk: Fecal immunochemical test (FIT) or laboratory evaluation; - Low risk: Screening may be deferred.	39.2%	Not reported	0.71–0.75	While the model improves risk stratification, it may require further validation across diverse populations to enhance generalizability.	Not yet FDA-approved or CE-marked; primarily utilized in research settings.	Used to expand CRC screening eligibility to younger populations based on risk.
b	ColonFlag™ [83]	Designed to identify individuals at high risk for CRC.	Developed using machine learning (random forests, decision trees) on age, sex, and 20 CBC parameters.	Used to stratify individuals at risk for CRC, assisting in early detection in asymptomatic patients and those who may not adhere to traditional screening programs.	25–48%	88–94%	0.80–0.82	The relatively low sensitivity suggests that ColonFlag may miss a significant number of CRC cases, limiting its effectiveness as a standalone screening tool.	CE-marked; FDA-exempt.	Supplementary tools for CRC screening to enhance early detection are used with FIT or colonoscopy.
b	Imperiale Model [84]	Designed to stratify risk for ACN in average-risk asymptomatic adults undergoing screening.	Developed using multivariable logistic regression on 13 variables: age, sex, marital status, education, smoking, significant alcohol use, metabolic syndrome, red meat consumption, aspirin/NSAID use, and physical activity.	Stratifies participants into low-, intermediate-, and high-risk groups for ACN to guide screening decisions.	Not reported	Not reported	0.58–0.62	The model developed on a predominantly white population, limiting generalizability.	Not yet FDA-approved or CE-marked.	Supports shared decision-making in CRC screening. Low-risk patients can opt for non-invasive screening tests like FIT; high-risk patients are recommended for colonoscopy.
b	Tao Model [85]	Designed to identify individuals at high risk for ACN among average-risk populations for targeted CRC screening.	Developed using logistic regression on 9 risk factors: sex, age, family history of CRC, smoking, alcohol consumption, red meat consumption, NSAID use, previous colonoscopy, and history of polyps.	Stratifies patients into risk categories (very low to very high) to prioritize screening colonoscopy for high-risk individuals and reduce unnecessary procedures for low-risk individuals.	Not reported	Not reported	0.65–0.69	Relies on self-reported risk factors, which may introduce inaccuracies. Focuses on ACN. May not be directly applicable to early-stage CRC detection.	Not yet FDA-approved or CE-marked.	Aids general practitioners and healthcare providers in identifying high-risk individuals for targeted screening, improving cost-effectiveness and compliance.
b	The Asia-Pacific Colorectal Screening Score Model [86]	To stratify risk for advanced colorectal neoplasia (ACN) in asymptomatic populations within the Asia-Pacific region.	It is a rule-based scoring system derived from logistic regression analysis using predefined demographic risk factors (age, sex, family history of CRC, and smoking status).	The tool identifies high-risk individuals for priority colonoscopy screening.	42%	86%	0.61–0.65	Limited sensitivity for detecting ACN. Validation is restricted to Asia-Pacific populations, limiting global generalizability. Does not encompass all risk factors for ACN (e.g., dietary habits, metabolic syndrome). High specificity but the potential for missing ACN in lower-risk categories.	Not yet FDA-approved or CE-marked.	Supports shared decision-making in CRC screening. Low-risk patients can opt for non-invasive screening tests; high-risk patients are recommended for colonoscopy. May conserve colonoscopy resources and improve screening uptake.
c	APAC [87]	Developed to improve surveillance of the at-risk population.	Age, sPDGFRβ, AFP, and creatinine.	Diagnosis of HCC in all stages of cirrhosis.	81.67%	95.35%	95	Tested only in patients with cirrhosis, no data for other “at-risk” populations (e.g., for NAFLD with bridging fibrosis, hepatitis B with a PAGE-B score > 10) is warranted.	No specific approval.	Tool for the identification of HCC, especially for early stages.
c	ASAP [88,89]	Online calculator of serum biomarkers to detect HCC among patients with chronic hepatitis B.	Age, gender, AFP, and PIVKA.	Diagnosis of HCC in the early stage of NAFDL.	82.7%	87.2%	89.8	Developed based on specific cohorts, and its accuracy might vary across different ethnicities, geographic regions, and healthcare settings.	No specific approval.	Diagnostic modes for detecting hepatocellular carcinoma (HCC).
				Diagnosis of HCC in early-stage CHB.	60.2%	90.4%	62.7			
c	HES [90]	Proposed to improve the performance of the serum alpha-fetoprotein (AFP) test in surveillance for HCC.	AFP, rate of AFP change, age, level of alanine aminotransferase, and platelet count.	Diagnosis of HCC in all stages of cirrhosis.	45.2%	90%	76	Primarily validated in specific cohorts (hepatitis C virus-related cirrhosis). Lack of data on different etiologies of liver disease or varying demographics.	No specific approval.	HCC early detection screening algorithm in chronic liver disease patients under surveillance for HCC.
c	GAAP [91]	Model for HCC detection.	Gender, age, AFP, and PIVKA-II.	Diagnosis of HCC in all stages of CLD.	87.2%	79.2%	91	Primarily developed and validated in Chinese cohorts, where hepatitis B virus (HBV) is a predominant cause of HCC.	No specific approval.	HCC detection in chronic liver disease population under surveillance for HCC.
c	BALAD 2 [92]	Statistical models for estimating the likelihood of the presence of hepatocellular carcinoma (HCC) in individual patients with chronic liver disease and the survival of patients with HCC, respectively.	Bilirubin, albumin, AFP-L3, AFP, and DCP.	Diagnosis of HCC in all stages of CLD.	87.7%	56.7%	89	Primarily developed for patients with cirrhosis, its predictive accuracy may not be reliable in patients without significant liver fibrosis or cirrhosis. This limits its broader application in the general population with liver disease.	No specific approval.	Diagnosis of HCC in individual patients with chronic liver disease and predicting patient survival.
c	GALAD [90,91,92,93,94,95,96]	Biomarker-based algorithm used to assess the risk of hepatocellular carcinoma (HCC) in patients with chronic liver disease.	Gender, age, AFP, PIVKA-II, and AFP L3.	Diagnosis of HCC in the early stage of CLD.	71–92%	73.493%	88–92	Heterogeneous data in different cohorts, different cutoffs evaluated, no standardization; majority in retrospective, case-controlled methodologies. The main limitations for implementation in clinical practice are the selection bias and the threshold values of these models for detecting early-stage HCC.	No specific approval.	HCC diagnostic in chronic liver disease patients under surveillance for HCC.
				Diagnosis of HCC all stages in CLD.	57.1–96.3%	79.9–95.7%	79–98			
				Diagnosis of HCC in all stages of NASH.	88.6%	95.3%	96			
				Diagnosis of HCC in the early stage of NAFLD.	77.8%	81.1%	87.4			
c	GAAD [97]	Aid in the diagnosis of hepatocellular carcinoma (HCC) (early and all stages).	Age, gender, AFP, and PIVKA II.	Diagnosis of HCC in the early stage of CLD.	70.1%	93.7%	91.4	Data available (case–control studies). Prospective cohorts are pending.	CE Mark, IVDR-approved.	Diagnosis of chronic liver disease and recommended for surveillance due to increased risk of developing HCC.
				Diagnosis of HCC all stages of CLD.	77.4–83.1%	89.6–93.7%	92–95			
d	FibroScan (Transient Elastography) [98,99,100,101,102]	Non-invasive assessment of liver stiffness as a surrogate for fibrosis.	Ultrasound elastography; liver stiffness measured in kilopascals (kPa).	Chronic liver diseases, including hepatitis B/C, NAFLD, and alcoholic liver disease.	70–90%	70–95%	0.80–0.91	Operator dependency; limited accuracy in obese patients and those with ascites.	FDA-approved.	Clinical practice for monitoring liver fibrosis and cirrhosis.
d	FibroTest (FibroSure in the USA) [103,104,105]	Non-invasive assessment of liver fibrosis and necroinflammatory activity.	Alpha-2-macroglobulin, haptoglobin, apolipoprotein A1, GGT, and total bilirubin.	Chronic hepatitis B/C, NAFLD, and alcoholic liver disease.	70–90%	70–90%	0.80–0.90	Affected by extrahepatic diseases, hemolysis, and Gilbert’s syndrome.	FDA-cleared; CE-marked.	Clinical practice for evaluating liver fibrosis and inflammation.
d	APRI (AST to Platelet Ratio Index) [106,107,108]	Predicting significant fibrosis and cirrhosis.	AST levels and platelet count.	Chronic hepatitis B/C.	60–80%	70–85%	0.70–0.85	Low sensitivity for early-stage fibrosis; less accurate in mild fibrosis; affected by thrombocytopenia.	Widely used in clinical practice; no specific approval.	Screening and initial assessment of liver fibrosis; predict fibrosis using routine blood tests.
d	FIB-4 (Fibrosis-4 Index) [109,110,111,112]	Predicting significant fibrosis.	Age, AST, ALT, and platelet count.	Chronic hepatitis B/C, HIV/HCV coinfection, and NAFLD.	65–85%	70–85%	0.75–0.85	Less accurate in mild fibrosis, affected by age and platelet count variations.	Widely used in clinical practice; no specific approval.	Screening and initial assessment of liver fibrosis.
d	Enhanced Liver Fibrosis (ELF) Test [113,114,115]	Non-invasive assessment of liver fibrosis.	Hyaluronic acid, TIMP-1, and PIIINP.	Chronic liver diseases, including hepatitis B/C, and NAFLD.	75–85%	75–85%	0.80–0.91	Affected by acute inflammation, limited data in certain populations.	CE-marked; FDA-approved.	Non-invasive fibrosis assessment using biomarkers.
d	Magnetic Resonance Elastography (MRE) [116,117,118,119]	Measure liver stiffness via MRI.	Liver stiffness measurement via MRI (MRI with low-frequency vibrations to assess liver elasticity).	Chronic liver diseases.	85–95%	85–95%	0.90–0.95	High cost, limited availability, contraindications in patients with metal implants.	FDA-approved.	Advanced imaging technique for assessing liver fibrosis and cirrhosis.
d	Acoustic Radiation Force Impulse (ARFI) Imaging	Assess liver stiffness with an ultrasound-based technique.	Ultrasound elastography measures shear wave velocity.	Chronic liver diseases, including hepatitis B/C, NAFLD.	70–90%	70–90%	0.80–0.90	Operator dependency, limited accuracy in obese patients.	FDA-cleared; CE-marked.	Ultrasound-based liver stiffness evaluation.
d	Shear Wave Elastography (SWE) [120,121,122,123,124]	Assess liver stiffness through ultrasound elastography.	Ultrasound elastography measures shear wave speed in liver tissue.	Chronic liver diseases, including hepatitis B/C and NAFLD.	75–90%	75–90%	0.80–0.90	Operator dependency; limited accuracy in obese patients.	FDA-cleared; CE-marked.	Clinical practice for evaluating liver fibrosis.
d	PROMETHEUS^®^ IBROSpect II [115,125,126,127]	Non-invasive assessment of liver fibrosis.	Hyaluronic acid, TIMP-1, and alpha-2-macroglobulin.	Chronic hepatitis C.	74%	74%	0.82	Limited data on other chronic liver diseases may be affected by acute inflammatory conditions.	FDA-approved.	Non-invasive fibrosis evaluation using biomarkers.
d	ADAPT Score [128]	Non-invasive assessment of liver fibrosis and cirrhosis.	Age, diabetes, and biomarkers.	Chronic hepatitis C.	70–85%	75–90%	82–88%	Limited applicability to viral hepatitis.	No specific approval.	Predict fibrosis risk in metabolic liver diseases.
e	Cardio Explorer (Exploris) [129]	Assessment of ACS/CAD patients.	32 clinical and lab parameters.	CAD patients.	82.3%	77.4%	0.87	Adaptation of laboratory testing panel in ED to include 15 lab values.	CE-marked.	Clinical evaluation of patients with suspected CAD to decide on further diagnostic modalities.
e	ACS Pathfinder (Artemis) [130]	Assessment of ACS patients.	Parameters: anamnesis, lab values, and ECG findings.	ACS patients in an emergency room setting.	Not reported	Not reported	0.95–0.98	No regulatory approval.	In the CE approval process.	Fast detection of NSTEMI patients with suspected MI based on a single and/or serial cardiac troponin measurement.
e	Chest Pain Triage Algo (Roche) [131]	Assessment of ACS patients.	Chest pain onset time and hsTn-values.	ACS patients in an emergency room setting.	Not reported	Not reported	Not reported	The algorithm uses only Roche hsTn.	In the CE approval process.	Aid in the interpretation of cardiac troponin results in the framework of validated European Society of Cardiology (ESC). 0/1 h, 0/2 h, and 0/3 h accelerated diagnostic algorithms for non-ST segment elevation. Myocardial infarction (NSTEMI) is based on a single and/or serial cardiac troponin measurement.
f	i-STAT TBI Plasma, Abbott [132]	Selecting patients by measuring the level of blood-based biomarkers associated with brain injury to determine the need for a head CT scan.	A panel of in vitro diagnostic immunoassays for the quantitative measurements of glial fibrillary acidic protein (GFAP) and ubiquitin carboxyl-terminal hydrolase L1 (UCH-L1) in plasma and a semi-quantitative interpretation of test results derived from these measurements. The test is to be used with plasma prepared from EDTA anticoagulated specimens in clinical laboratory settings by a healthcare professional and is not intended to be used in point-of-care settings.	Test results are interpreted with other clinical information to aid in evaluating patients who have suffered mild TBI (Glasgow Coma Scale 13–15) within 12 h of injury.	95.8%	40.4%	Not reported	Timing restriction because blood-based biomarkers associated with brain injury peak most commonly within 24 hafter mTBI. Not for pediatric use.	FDA-approved; CE-marked	The test measures the level of biomarkers associated with brain injury in the bloodstream to assist in determining whether a CT scan of the head is needed in patients 18 years of age or older (II).
f	Brain Trauma Assessment Kit, Banyan Biomarkers Inc. [132,133]	The Brain Injury Screening Tool (BIST) helps health practitioners assess and manage patients after brain injury (or concussion) who are over 8 years old.	Follow the prompts and enter patient responses. Anamnestic and clinical parameters.	The first symptoms checklist is designed to assist in identifying patients at the following risk levels: - Moderate/high risk of poor recovery or who need specialist referral; - Low-risk patients who are likely to recover well and can be supported with ongoing monitoring and advice, usually with no need for neuroimaging and hospitalization.	Not reported	Not reported	Not reported	Operator dependency, which is based primarily on patient answers. Limited to subjects who are over 8 years old.	User guide; no specific approval,	To identify clinical indicators that the patient is at significant risk of long-term neurological disabilities (III).
f	EMATS [134]	Assessing prognosis of TBI and stroke.	EEG-based Machine or Deep Learning Algorithm for TBI Stroke Classification (EMATS) through the use of the included EEG preprocessing, feature extraction code, and machine learning models that have been trained on a large dataset.	Non-invasive support together with clinical assessment of medical devices where classification of resting EEG signals is needed (“Normal”, “TBI”, “Stroke”).	Not reported	Not reported	Larger than 0.76	Set of machine or deep learning algorithms but not a diagnostic tool; EEG availability; limited to subjects who are between 18 and 65 years and do not have any previous history of epilepsy; no information about phase and severity of mTBI or stroke.	Available large patient data set, no specific approval.	EEG-based machine or deep learning algorithm is used to assess the prognosis of TBI and stroke (III).
f	Resting-state Functional Network Connectivity (rsFNC), The Mind Research Network [135]	Adequate detection of mTBI and risk stratification for long-term psychiatric, neurologic, and psychosocial disabilities.	Detection of mTBI by machine learning classification using resting state functional network connectivity and fractional anisotropy.	A promising method to collect unique information to detect mTBI and prevent long-term neurological disabilities properly.	84.1%	Not reported	Not reported	A small number of samples, a relatively simple method used for feature selection, research-based and not commercial.	No specific approval.	A promising option for the diagnosis of mTBI (II). Non-invasive risk stratification and prevention of long-term psychiatric, neurologic, and psychosocial problems (III).
f	Artificial Neural Network (ANN) Model, Queensland Brain Institute [136]	Identifying positive mTBI from negative mTBI subjects.	Two machine learning (ML) models to diagnose mTBI in a pediatric population were collected as part of the Paediatric Emergency Care Applied Research Network (PECARN) study. The models were conducted using patients under the age of 18 years with mTBI and had a head CT report. In the conventional model, random forest (RF) ranked the features to reduce data dimensionality, and the top-ranked features were used to train a shallow artificial neural network (ANN) model. In the second model, a deep ANN is applied to classify positive and negative mTBI patients using the entirety of the features available.	To diagnose mTBI in a pediatric population (II), identifying positive mTBI from negative mTBI patients.	99.2%	99.5%	Not reported	Limited to the pediatric population, research-based, and not commercial.	No specific approval.	The detection of mTBI in pediatrics using deep ANN through clinical and non-imaging data. The diagnosis of mTBI is therefore performed with balanced sensitivity and specificity using shallow and deep machine learning models.
f	PECARN (Pediatric) Emergency Care Applied Research Network) [137]	Identify pediatrics to assist CT decision-making after mTBI.	Clinically important TBI (ciTBI) was chosen as the primary outcome because it is clinically driven and accounts for CT’s imperfect test characteristics. In the less than 2-year-old group, the rule was 100% sensitive. In the greater than 2-year-old group, the rule had 96.8% sensitivity. In those under 2 with Glasgow Coma Scale (GCS) = 14, Altered Mental Status (AMS), or palpable skull fracture, the risk was 4.4%, and CT imaging is recommended. (Risk with any of the remaining predictors was 0.9% and less than 0.02% with no predictors.) In those over 2 with GCS = 14, AMS, or signs of basilar skull fracture, the risk was 4.3%, and CT imaging is recommended. (Risk with any remaining four predictors was 0.9% and less than 0.05% with no predictors.) PECARN prediction rule outperformed both the CHALICE and the CATCH clinical decision aids in external validation studies.	Age-based PECARN TBI prediction rules to accurately identify children at very low risk for a clinically significant TBI that can be used to assist CT decision-making for children with minor blunt head trauma (II).	100%	69.9%	Not reported	Limited to the pediatric population.	External validation through clinical trials.	The PECARN Pediatric Head Injury Prediction Rule is a clinical decision aid that allows physicians to safely rule out the presence of clinically important traumatic brain injuries, including those that would require neurosurgical intervention among pediatric head injury patients who meet its criteria without the need for CT imaging.
g	Sepsis ImmunoScore (Prenosis) [138,139]	Identification of patients at risk for having or developing sepsis within 24 h.	Up to 22 parameters, including vitals, demographics, CBC-, BMP/CMP panel tests, and 3 lab tests (PCT, CRP, and lactate).	Risk stratification of ED patients suspected of having sepsis.	Not reported	Not reported	0.81	For adults only (18 years old); need of ordered blood culture.	US FDA de novo clearance.	Aid in diagnosis of ED patients suspected of having sepsis. Active ordering of the test.
g	COMPOSER (UCSD) [140,141]	Prediction of onset of sepsis 4–48 h prior to time of clinical suspicion.	40 clinical variables, of which 34 were dynamic and 6 were demographic.	Early sepsis prediction in ED and ICU.	90.5% (in ED)	94.7% (in ED)	0.938 (in ED)	For adults only (18 years old).	Research-based; no specific approval.	Continuous patient surveillance for signs of possible sepsis.
g	TREWS (Bayesian Health) [142,143]	Risk prediction for septic shock with a median lead time of 24 h.	Up to 54 features derived from routinely available measurements in the EHR.	Detection of patients at high risk of developing septic shock.	85%	67%	0.83	For adults only (18 years old).	No specific approval.	Continuous patient monitoring for risk of sepsis (Early Warning System).
g	Epic Sepsis Model [144]	Real-time risk prediction of sepsis automatically calculated every 20 min.	Various parameters, including demographic, comorbidity, vital signs, laboratory, medication, and procedural variables.	Prediction of a patient’s risk of sepsis at a given point in time.	Not reported	Not reported	0.76–0.83	For adults only (18 years old).	No specific approval.	Continuous patient monitoring for risk of sepsis (Early Warning System).
g	NAVOY^®^ CDS (AlgoDx) [145]	Automatic qSOFA score calculation indicates patients with suspected infection who are at greater risk for a poor outcome.	20 clinical parameters, including vital signs, blood gas tests, lab values, gender, and age.	Detection of patients who are likely to be septic.	80%	78%	0.80	For adults only (18 years old).	CE-marked and US FDA 510(k) clearance	Automatic qSOFA score calculation of suspected septic patients.
g	Sepsis Sniffer (Mayo Clinic) [146]	Automated surveillance algorithm for the detection of severe sepsis and monitoring failure to recognize and treat severe sepsis in a timely manner.	9 pathophysiologic variables.	Prediction of a patient’s risk of sepsis at a given point in time.	80%	96%	0.96	For adults only (18 years old).	Research-based; no specific approval.	Continuous patient surveillance for risk of sepsis (Early Warning System).
h	KFRE(4-Variable Version) [147,148]	A quantitative risk score predicts the likelihood of an individual patient to reach end-stage kidney disease within 2 and 5 years. This is a publicly available regression-based risk score. It can be applied to diagnosed patients with chronic kidney disease (CKD) in stages G3 to G5.	Urine, sex, age, and GFR.	Chronic kidney disease.	Depending on the cut-off; not provided in this paper	Depending on the cut-off; not provided in this paper	0.80 at 2 years and 0.77 at 5 years	Developed for use in stages 3 to 5, not for earlier stages.	Widely known and used in clinical practice; no specific approval for the publicly available version.	Aid in assessment of the risk of progression of end-stage CKD.
h	KPNW [149]	Risk assessment for progression to kidney failure in 5 years.	8 variables, incl. age, sex, diabetes status, diabetes complications severity index, mean systolic BP, antihypertensive medication use, eGFR, hemoglobin, and proteinuria/albuminuria.	Chronic kidney disease.	92.2% using the top quintile of predicted risk as the cutoff	Not disclosed	0.95	Developed for use only for stage 3 or 4 CKD, with limited external validation.	No specific approval.	Risk assessment for progression to kidney failure.
h	KidneyIntelX. Dkd [150]	Aid in assessment of the risk of progressive decline in kidney function (sustained decrease in eGFR greater than or equal to 40% lasting more than 3 months) within up to 5 years following KidneyIntelX.dkd level measurement in adult patients with type 2 diabetes and existing chronic kidney disease.	K2EDTA plasma TNFR1, TNFR2, and KIM-1 and clinical data.	Chronic kidney disease.	Depending on the cut-off; not disclosed	Depending on the cut-off; not disclosed	0.777	Developed for use only for diabetic kidney disease with limited external validation; the sample size of development and validation is relatively low.	FDA-cleared.	Aid in assessment of the risk of progressive decline in kidney function in adult patients with type 2 diabetes and existing chronic kidney disease.
h	The Klinrisk Model [151]	Aid in the assessment of the risk of progressive decline in kidney function and/or reaching end-stage kidney disease within a period of up to 5 years in adult patients diagnosed with chronic kidney disease (CKD) stages G1 to G4 and adult patients at risk.	Age, sex, eGFR, and urine ACR, and an additional 18 laboratory results from chemistry panels, liver enzymes, and complete blood cell count panels.	Chronic kidney disease.	2 years:For low risk (lowest 50% population), 0.91 sensitivity For high risk (top 10% of the population), 0.65 sensitivity	2-year:For low risk (lowest 50% population), 0.51 specificity For high risk (top 10% of the population), 0.91 specificity	0.87 (0.86–0.88) at 2 years	No regulatory approval limits its clinical use; real-world effectiveness and generalizability in other markets/populations are to be demonstrated.	No approval from the health authority.	Aid in assessing the risk of progressive decline in kidney function and/or reaching end-stage kidney disease within up to 5 years.

#### 2.1.1. Lung Cancer

One of the primary challenges in cancer screening programs is the failure to identify high-risk individuals. Current guidelines for lung cancer screening recommend using a categorical approach based on demographics, smoking status, and number of pack years as standard criteria to determine eligibility for annual low-dose computed tomography (LDCT) [152]. While sensitivity and specificity are critical metrics of screening effectiveness, LDCT excels in identifying true positive cases of lung cancer but suffers from low specificity. This limitation leads to a high rate of false positives and overdiagnosis, resulting in unnecessary tests and invasive procedures [153]. It is estimated that 18% to 67% of lung cancers detected through LDCT may be overdiagnosed, which exposes patients to unnecessary risks [154]. Another emerging concern is the rise in lung cancer rates among nonsmokers, a group of individuals that remains largely excluded from screening recommendations. These observations indicate the need for more inclusive approaches [155]. Improving the identification of high-risk patients is crucial for increasing the benefits of lung cancer screening while minimizing its harm to the lower-risk group. Early models (Table 3a) focused on simple risk factors such as demographics, smoking status, and occupational exposure; the most recent one included a broader range of risk factors, including comorbidities, family cancer history, and more variables on smoking behavior like intensity and duration. These models have been proven effective given their higher discriminative power than the age–smoking history categorical approach [156]. The introduction of AI and ML learning led to a significant advancement in the development of risk prediction models.

LungFlag™ is one notable model that uses ML to assess lung cancer risk by analyzing novel predictors like white blood cells and platelets in addition to the established parameters. An evaluation among never-smokers demonstrated that the ML model was more accurate than standard eligibility criteria for lung cancer screening in identifying lung cancer as early as 9–12 months before clinical diagnosis. Interim findings from the prospective international lung screening trial showed that the use of the PLCOm2012 model had been associated with higher cancer detection rates and greater cumulative life expectancies at 6 years compared to the traditional approach [80]. From a cost perspective, Toumazis et al. also demonstrated that risk model-based screening strategies with a 6-year risk threshold of 1.2% or greater were more cost-effective than the categorical approach with an incremental cost-effectiveness ratio of less than USD 100,000 per quality-adjusted life-year (QALY) [157]. These findings indicate that prioritizing high-risk individuals is beneficial in lung cancer screening.

#### 2.1.2. Colorectal Cancer

Besides the advantages of identifying high-risk individuals, low compliance rates (CR) might impair cancer screening effectiveness. CR is a metric used for the evaluation of screening programs [158] and is often recognized as a major challenge [159]. A notable example is CRC screening, which uses a one-size-fits-all approach based on a single age-based criterion to initiate screening. CRC screening is a two-step-process:Fecal immunochemical test (FIT) as a primary modality;Colonoscopy is recommended for individuals with positive FIT results.

Because only 25% to 55% of individuals with positive FITs receive a follow-up colonoscopy and evaluation [160], the risk of death is 2-fold higher among non-compliers [161]. The low compliance rate is primarily due to the colonoscopy’s invasiveness and higher complication rates [162]. Several risk models are available to address this inefficiency, incorporating additional risk factors for stratification, such as sex, family history, body mass index, smoking, and alcohol intake [163].

These models may address the challenges of CRC population-based screening programs by differentiating high-risk individuals who should undergo colonoscopy and individuals with average/low risk that could be screened with noninvasive methods such as FITs, thereby reducing the demand for colonoscopies [164]. The well-known Asia-Pacific Risk Score (APCS), validated across 15 Asia-Pacific countries, best exemplifies this benefit. When combined with the FIT, the APCS can detect about 70% of subjects with advanced neoplasia and 95% with cancers. This combination significantly outperforms FIT screening alone and reduces the demand for direct colonoscopy screenings by half [165]. A head-to-head comparison between APCS and over 16 CRC risk models using more or less similar risk factors showed that no single risk score model was proven to be better than the other. All risk models exhibited variable but generally modest predictive performance, with area under the curve (AUC) values ranging from 0.57 to 0.65 [163]. Most models in Table 3b demonstrate moderate sensitivity (identifying individuals with CRC or advanced neoplasia correctly) and specificity (correctly identifying those without CRC), indicating room for improvement. For example, APCS has a sensitivity of 42% and specificity of 86%, which is sufficient to reduce unnecessary colonoscopies but may miss a significant portion of high-risk individuals.

ColonFlag™, a novel AI-driven risk prediction model, (provided by Medial Early Sign, Hod Hasharon, Israel), addresses some of the limitations of traditional risk scores. Unlike models that rely only on risk factors like age, sex, family history, and lifestyle, ColonFlag™ uses a complete blood count. A retrospective observational study conducted in two independent datasets showed that ColonFlag™ outperformed the traditional screening methods in identifying people at 10 to 30 times increased risk of CRC. In large datasets of 75,822 records, ColonFlag™ was able to identify up to 48% more CRC cases than the guaiac fecal occult blood test (gFOBT) [70] with an AUC of 0.80–0.82, indicating good discriminative power. ColonFlag^®^ was validated across multiple healthcare systems, including Israel, the UK, and the US, with 88–94% specificity rates, making it a valuable tool in cancer screening programs [165].

While CRC risk score models offer valuable tools for improving the effectiveness of population-based screening, their modest predictive performance underscores the need for innovation, such as the hybrid approach of combining these models with standard-of-care methods like FIT [166]. However, further research should focus on improving sensitivity to minimize missing advanced neoplasia or CRC while maintaining high specificity to avoid unnecessary procedures.

Additionally, studies on CRC screening risk models are limited; thus, evidence for the financial benefits is still inconclusive [167]. This emphasizes the need for further validation research to showcase clinical and economic value across diverse populations and healthcare settings.

#### 2.1.3. Hepatocellular Carcinoma

Hepatocellular carcinoma (HCC) has a 5-year survival rate persistently below 20%. The prognosis varies on tumor staging at diagnosis, with an early-stage median survival of 5–10 years [168]. HCC is the sixth most prevalent cancer, with over 860,000 new cases annually, and the third most deadly, causing over 750,000 deaths in 2022 [169]. HCC death rates rise by 2–3% annually due to late diagnosis and lack of curative therapies for advanced stages. Liver cirrhosis, present in 70–90% of cases, is a major risk factor, with hepatitis B (HBV) and C (HCV) infections accounting for over 50% and 25–31% of cases. Other risk factors, such as non-alcoholic fatty liver disease (NAFLD) and non-alcoholic steatohepatitis (NASH), are increasingly linked to HCC, with significant fractions found in non-cirrhotic livers [170]. A meta-analysis of 59 studies, including 145,396 HCC patients, found that surveillance is associated with improved early-stage detection, curative treatment, and overall survival [171]. All clinical guidelines recommend surveillance. Research focuses on advancing imaging techniques and identifying non-invasive biomarkers for detection accuracy, such as alpha-fetoprotein (AFP), the most commonly used add-on serum marker, which remains suboptimal. Other challenges in clinical practice include inadequate risk stratification and underuse of surveillance. Within HCC, AI models are being incorporated into the radiological and histological detection of the disease. Multi-marker models and AI-driven approaches are expected to improve the sensitivity and specificity of HCC surveillance, making it possible to identify at-risk individuals more accurately [172]. Several algorithms that combine various biomarkers with clinical features have been developed and tested. However, for HCC detection, most of these (with one exception) remain research tools, as they have not received regulatory approval due to the heterogeneous results from clinical performance studies, which have been conducted in different populations and with varying cutoff values (Table 3c). As for liver fibrosis evaluation, several algorithms have already been registered (CE mark or FDA approval), proving the value of this approach (Table 3d).

Accurate diagnosis of liver fibrosis is essential for managing and preventing chronic liver diseases (CLD). Liver fibrosis can progress to cirrhosis and HCC if left undetected. This progression represents a significant global health burden, affecting over 1.5 billion people worldwide, with NAFLD affecting approximately 25% of the global population due to rising obesity and type 2 diabetes rates [173]. Early and precise detection is critical, enabling timely interventions, which can halt disease progression and improve patient outcomes. Current diagnostic methods include non-invasive tests such as transient elastography (FibroScan), blood-based biomarkers (FibroTest), and imaging techniques like Magnetic Resonance Elastography (MRE) and Acoustic Radiation Force Impulse (ARFI) imaging. While these methods are valuable, they have notable limitations. Liver biopsy, though a gold standard, is invasive and may not fully capture the heterogeneity of fibrosis. Non-invasive imaging methods, including elastography and MRI-based techniques, offer less risk but may lack precision, especially in detecting early-stage fibrosis. Next to this, costs remain a restrainer when applying these methods [107]. However, advancements in AI and ML present promising solutions for accurately assessing fibrosis. AI models often surpass traditional methods in sensitivity and specificity, enhancing diagnostic accuracy, accessibility, and cost-effectiveness [174].

### 2.2. Coronary Artery Disease

Early diagnosis of non-ST-segment elevation myocardial infarction (NSTEMI) is critical. However, even if myocardial infarction is not yet confirmed, an obstructive CAD is a potential pending threat, keeping in mind that a non-negligible number of patients with heart attacks miming obstructive CAD are caused instead by non-obstructive conditions such as coronary artery spasms. For the diagnosis of “myocardial infarction with non-obstructive coronary arteries” (MINOCA), elevated myocardial necrosis biomarkers play a crucial role as well [175]. To restore blood flow to the damaged portion of the heart is an absolute priority. The combination of two CDS systems in a consecutive manner shall highlight how AI may help solve this unmet medical need: Highly sensitive cardiac Troponin T (hs-cTnT) tests have made a huge difference in the time needed to diagnose NSTEMI. Nevertheless, the entire workflow of ruling patients in/out is time-consuming, especially for patients with diabetes who lack symptoms [176]. Rapid diagnostic protocols, like the ESC 0/1, ESC 0/2, and ESC 0/3, address this topic [177]. Created in collaboration with the University Hospital Heidelberg, the chest pain triage algorithm (CPTA) is a CDS tool designed to improve the diagnostic procedure for patients with non-ST-segment elevation myocardial infarction (NSTEMI). Conventionally, diagnosing NSTEMI requires, on average, 6–7 h; in 25% of chest pain patients, more than two blood samples are needed [178]. The application of the ESC 0/1 h protocols significantly decreased the length of stay in the ED to 4.5 h [178]. The CPTA for high-sensitivity Troponin T (hs-cTnT) enables the automatic application of the appropriate ESC rapid rule out/in the protocol, and fosters change towards implementing the ESC 0/1 h protocols.

Patients ruled out for NSTEMI may be assessed with Cardio Explorer, a CDS solution from Exploris [179], for their potential risk of CAD. This AI system examines 32 predictive factors, offering more accurate risk stratification than the conventional approach, which is the Pre-Test Probability (PTP) score. Recent research showed that Cardio Explorer (provided by Exploris Health AG, Wallisellen, Switzerland)accurately categorized 67.7% of outpatients with low to intermediate risk as “very low risk” and 8.5% in the very high-risk class recommending direct invasive coronary angiography vs. 7.0% and 0%, respectively, with the PTP score [180]. These novel CAD algorithms demonstrate high accuracy (AUC 0.87 and 0.95–0.98) and strong potential for early cardiac risk identification and fostering faster life-saving interventions(Table 3e).

### 2.3. Traumatic Brain Injury

TBI is a substantial public health burden with extremely high incidence [181] and debilitating neurological outcomes after acute events. Around 50% of adult patients with mild TBI presenting to the hospital do not recover to pre-TBI levels of health by 6 months after their injury [182,183], making tertiary prevention essential for the patient. Although low- and mild-energy TBI, defined as a Glasgow Coma score of 13–15, comprises most TBI cases (over 90%) presenting to the hospital, fewer than 10% of patients discharged after presenting to an emergency department for TBI in Western countries receive follow-up [181,182].

According to several hospital protocols, CT scans are conducted and assessed for lesions triggering hospital admission or life-saving surgery. However, those triage protocols are inefficient for patients after low- and mild-energy TBI, with more than 90% of scanned patients showing no intracranial finding but being exposed to radiation risk. The InTBIR Clinical Studies [184] have demonstrated that measurement of blood-based biomarkers such as S100B, GFAP, UCH-L1, Tau, and NF-L [185] adds substantial value to clinical decision rules, holding the potential to improve efficiency while reducing radiation exposure. Low and mild TBI are still showing the highest false-negative rate (FNR) and readmission diagnoses after discharge from the ED [186]. Despite no findings at CT scans, increased concentrations of biomarkers in patients’ blood suggest structural brain damage visible on MRI scanning in up to 30% of patients with mild TBI [181,185]. Thus, secondary prevention is key to pushing scientific research and commercial companies to find innovative solutions such as i-STAT TBI Plasma^®^ (provided by Abbott Point of Care Diagnostics, Princeton, NY, USA)). Nevertheless, it is not without limitations. Although their commercially available and validated model (Table 3f) shows high sensitivity equal to 95.8%, the relatively low specificity not exceeding 40,4% addresses the potential need to perform the best neuroimaging such as MRI to confirm parenchymal changes, correctly suggested by increased levels of blood-based biomarkers and not detected by CT scan.

Development efforts should focus on supporting physicians in the ED by identifying patients after low and mild TBI who may be discharged safely, omitting imaging diagnostics, and presenting persisting low levels of blood-based biomarkers. Such algorithms may also identify patients who, despite initial CT scan without findings, will benefit from further imaging, such as an MRI scan, to verify potential peri-axonal swelling or bleeding, invisible at the CT scan but correctly suggested by increased blood-based biomarkers.

At this point, the most promising diagnostic and risk-stratification tools are only available as research-based large patient data sets; only a few are commercially available and validated. Nevertheless, all those development efforts (Table 3f) emphasize the necessity to impact healthcare systems economically and viably.

### 2.4. Sepsis

Sepsis is a major global healthcare threat associated with severe disease burden, including high economic costs and health outcomes impact. In the US, sepsis treatment costs more than USD 41.5B annually [187]. It is a leading cause of unplanned readmissions, causing more than USD 3.5 billion in estimated annual costs in the US alone [188]. Globally, it is causing approximately 11 million deaths each year [64] (nearly 1 in 5 of all annual deaths), surpassing the combined total of all cancers, which account for 9.7 million deaths [58].

Patients with suspected sepsis are frequently diagnosed late [189]. Each hour of delay in appropriate antimicrobial administration reduces the chance of survival by 7.6% [190]. Hence, there is a critical need for screening technologies to reliably identify high-risk patients at home and in the hospital. Furthermore, the reliable discrimination between viral and bacterial infections remains an important puzzle piece in the diagnostic arsenal [191]. For bacterial infections, 20–30% of sepsis patients [192] do not receive effective antibiotic treatment promptly, which underscores the necessity to establish targeted antibiotic therapy within 1 h [193].

Additionally, biomarkers and clinical parameters should accurately predict the disease course to enable the timely implementation of supportive measures. This includes algorithms that suggest escalation instances, such as transferring a patient from the ward to the intensive care unit and vice versa. Prenosis Sepsis ImmunoScore, for example, is the first FDA-authorized AI-based algorithm designed to aid in the identification of patients at risk for either having or developing sepsis [138]

The algorithm can also predict key adverse outcomes, including in-hospital mortality, length of hospital stay, intensive care unit (ICU) admission, mechanical ventilation, and vasopressor administration within 24 h [138]. The CDS tool utilizes up to 22 parameters, such as vital signs, demographics, and clinical and sepsis-related laboratory biomarkers, to generate a risk score and assign the patient to one of four distinct risk stratification categories. Sepsis ImmunoScore integrates into a hospital’s EMR, allowing seamless incorporation into the regular clinical workflow [138].

Although the models listed in Table 3g demonstrate robustness in their original studies, a growing body of literature suggests that their actual performance in clinical practice may be lower than what was reported by the original developers [194]. Additionally, various healthcare organizations (HCOs) might have different protocols for sepsis diagnosis and management (even differing definitions of sepsis itself), further complicating the translation of these original studies into real-world settings.

The space of sepsis algorithms is shifting from purely Early Warning Systems (mostly passive automated models) to fully regulated software such as medical devices (SaMDs). The increased regulatory requirements also necessitate post-marketing evidence generation to ensure that these models are correctly and beneficially implemented within clinical workflows, thereby truly supporting healthcare providers in diagnosing sepsis and eventually improving patient outcomes [195]. This shift necessitates that the models become more accurate, aiming to increase the true-positive rate and significantly decrease the false-negative rate, a major contributor to alert fatigue, particularly in sepsis diagnosis [196].

### 2.5. Chronic Kidney Disease

CKD is an escalating global health burden, affecting 9.1–13.1% of the world’s population [64]. Prevalence varies, with most affected individuals being asymptomatic and underdiagnosed [197]. Even asymptomatic or early CKD is associated with an elevated cardiovascular (CV) risk. The impact of CKD on healthcare systems is substantial, accounting for 2–4% of national healthcare expenditures in developed countries for treating end-stage CKD (ESKD). In Europe, these costs are estimated to be around €140 billion annually [198,199]. The KDIGO clinical practice guidelines recommend using clinically validated risk prediction algorithms, such as the kidney failure risk equation (KFRE) [199]. The four-variable KFRE provides a percentage risk of ESKD within a specified time. Validated in over one million patients, its use is recommended by various national and international guidelines [200,201] to inform treatment decisions and monitoring pathways. Developed for use in CKD stages 3–5, the model utilizes age, gender, eGFR, and uACR, enabling the calculation of the percentage risk of progression to ESKD within 2 and 5 years, respectively. The algorithm equips clinicians to identify patients at low risk, focusing on cardiovascular disease prevention, and those at higher risk who require cardiovascular disease prevention and referral to secondary care. KFRE has been validated in more than 30 countries worldwide, making it the most validated algorithm for determining a patient’s risk of end-stage KD (Table 3h). It is included in the NICE guidelines [202] to aid risk assessment and referral decisions. However, this algorithm was developed for use in CKD stages 3 to 5 and not in earlier stages. A couple of novel algorithms are developed to identify patients at high risk of CKD for early intervention, thereby slowing down disease and downstream morbidity (Table 3h).

A machine learning tool, the Klinrisk model, was developed to predict the risk of CKD progression at all stages of the disease [203]. The model was developed in a population-based cohort from Canada, with external validation in another cohort leveraging routine laboratory data to predict eGFR decline or kidney failure. KidneyIntelX.dkd was developed and validated to assess the risk of progressive kidney function decline in adults with diabetes and early-stage kidney disease and received FDA de novo marketing authorization in 2023 (Table 3h). Overall, these novel algorithms have moderate to high accuracy (AUC above 0.78) and have great potential to identify patients at high risk at an early stage for early intervention. However, there is limited external validation in other markets and populations and limited evidence of their effectiveness in real-world clinical practice.

#### Summary: Algorithms in Prevention

In lung and CRC, AI-developed models like LungFlag™ and ColonFlag™ have demonstrated improved risk prediction, early detection, and cost-effectiveness compared to standard screening methods. Also, AI tools for fibrosis offer a promising alternative to liver biopsies for assessing liver disease progression. Some of these algorithms have received FDA approval and CE marking, indicating they meet regulatory clinical use standards. However, their effectiveness can still be variable, as performance often depends on factors such as the underlying liver disease, patient demographics, and the specific biomarkers incorporated. Novel CAD algorithms show high accuracy and offer a faster cardiac risk detection approach, impacting the clinical workflow in early diagnosis of non-ST segment elevation myocardial infarction (NSTEMI) in critical care. Their usage will result in faster decision-making on life-saving interventions, such as percutaneous coronary intervention, and offer superior workflows compared to the standard of care. In TBI, AI models exist that are able to select patients who will benefit from further imaging diagnostics and preventive neurorehabilitation while using available resources more efficiently. Algorithms in sepsis can abbreviate time to diagnosis and diagnostic accuracy. The development of such algorithms would greatly benefit from a consensus on the precise definition and management of sepsis. Until such an agreement is reached, it remains difficult to make definitive statements about their performance. Finally, available algorithms in CKD reported superior performance compared to the standard of care, e.g., KDIGO heatmap, in the development and validation cohorts. Despite the potential of AI-based algorithms to innovate healthcare, more research is required to improve generalizability and predictive power and to confirm clinical and economic unity across various populations and healthcare systems. Real-world validation, standardized protocols, and broader testing are required to expand the use of those tools in secondary prevention. In summary, AI-based algorithms play an increasing role in early detection and screening by enhancing risk stratification, especially in oncology. In tertiary prevention, algorithms help manage diseases and prevent complications in conditions such as acute coronary care, traumatic brain injuries, and sepsis.

## Figures and Tables

**Table 2 diagnostics-15-00648-t002:** Critical features and corresponding evaluation metrics to identify clinically meaningful algorithms.

Feature	Explanation	Sample Metrics
Explainable	CDS systems should provide clear, understandable rationales for their recommendations. Explainability ensures that users can comprehend how and why a decision was made, enhancing trust and enabling them to justify the recommendation to patients or peers. The algorithm’s developer should precisely define influencing factors and input parameters in their quantity and causality to make the result understandable. Algorithms could include interpretable models or visualization tools that translate complex computations. Several metrics are recommended to evaluate the explainability: faithfulness, robustness, clinical relevance, understandability, plausibility, etc.	Faithfulness (the correlation between input parameter importance scores and their actual impact on predictions); Robustness (stability of the explanation due to minor input changes); Clinical relevance (correlation of the input parameters to existing medical literature and guidance); Understandability (human-interpretable reasons for its predictions) [43]; Plausibility (the agreement between algorithm-generated explanation and human-annotated ground truth).
Accurate	Accuracy is paramount for CDS, as errors can have serious consequences. Quality metrics should include sensitivity, specificity, and predictive value measures, ensuring the system can reliably identify conditions, recommend treatments, or predict outcomes. Continuous benchmarking against gold-standard datasets and clinical outcomes is essential.	Traditional metrics to ensure clinical accuracy and effectiveness are sensitivity, specificity, area under the curve (AUC) and the Concordance Index (C-index), Positive Predictive Value (PPV), and Negative Predictive Value (NPV).
Dynamic	In clinical practice, patient conditions and medical contexts are constantly changing. A dynamic algorithm can incorporate new information as it becomes available (such as patient vitals, laboratory results, or imaging findings), enabling it to make updated and accurate predictions—the ability of the algorithm to make dynamic predictions using new data as they become available.	Consistency of accuracy over time: The predictive accuracy score (e.g., area under the receiver operating characteristic curve [AUROC]) is measured periodically as new data streams emerge. Error reduction with sequential data: Percentage reduction in prediction error when integrating additional data compared to static models.
Autonomous	The algorithm’s ability to generate automated results without involving the end user is balanced with human oversight to prevent overreliance. Quality metrics should evaluate the CDS system’s ability to operate autonomously while allowing users to review, confirm, or override recommendations when needed.	Effective measures include the percentage of time saved, the percentage of user error reduction, the percentage of system error reduction, etc. Error detection and notification: Error detection precision (e.g., percentage of flagged cases requiring user intervention).
Fair	Quality of the algorithm to objectify influencing parameters and to prevent selective bias and inequity (e.g., vital parameters compared to subjective anamnestic data).	Test fairness (performance across various populations, such as age, gender, ethnicity, and socioeconomic status); Cross-group ranking (whether the system’s rankings are consistent across different demographic groups); Equalized odds and opportunity (system has equal false positive, false negative, and true positive rates across different demographic groups); Bias mitigation (methods used to reduce unfair discrimination in the system’s outputs) [44].
Reproducible	To ensure the traceability of the algorithmic decision-making process, algorithm quality must be validated prospectively or retrospectively in independent cohorts. Thus, a CDS algorithm must demonstrate consistent performance across different clinical settings and datasets.	Technical reproducibility (if the same results can be obtained using identical tools and datasets); Statistical reproducibility (if the variance around the results is reported and consistent); Conceptual reproducibility (if the desired outcome can be replicated under different clinical settings) [45].
Medical Value	The ability of an algorithm to address an unmet medical need.	Actionable insights that lead to improved patient outcomes.
User-Centric Design	A CDS system’s usability should be rigorously tested to ensure it fits naturally into the end user’s workflow.	Metrics should evaluate factors like interface intuitiveness, time-to-task completion, user satisfaction, error rates (number and severity), user confidence, learnability, etc.
Data Privacy and Security	Robust data protection mechanisms should safeguard sensitive patient information. Compliance metrics, such as HIPAA and GDPR compliance and adherence to industry standards for encryption and data access control, are critical.	Common metrics include encryption strength, vulnerability assessment frequency, audit trail comprehensiveness, data backup and recovery efficiency, compliance audit performance, data anonymization rate, and reidentification risk assessments.

## Data Availability

No new data were created or analyzed in this study. Data sharing is not applicable to this article.

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
