# Peer review of "The Value of Clinical Decision Support in Healthcare: A Focus on Screening and Early Detection"

_diagnostics, 2025, doi:10.3390/diagnostics15050648_

Round 1
Reviewer 1 Report
Comments and Suggestions for Authors
HH Schäfer et al. conducted a review article aims to describe the purpose and mechanisms of clinical decision support systems (CDSS), to discuss different entities of algorithms, to assess quality features, to describe contemporary algorithms in Oncology, Acute Care Cardiology and Nephrology and to discuss the challenges and limitations of CDSS in clinical practice. The authors did a great work but there is a need to clarify some points detailed below:
Major comments
- The first 3 chapters are generalities that can be regrouped in one. The conclusion of this first part could introduce the next one.
- In the chapter “Algorithm types of CDSS through the lens of prevention”, a text explaining why this is the focus of the manuscript could be helpful, as well as an explanation on the methodology used to select the CDSS presented and the reason why choosing oncology, acute care and nephrology (include also a small sentence on that in the abstract and summary).
- In the Table S2 to S10 that would be helpful to have the same variables presented including “input data” and similar variables as in Table 6 (“Limitations”, “Regulatory Status”, “Clinical Use”).
- Chapter 6 (“Challenges and limitations of CDS”) is very short, should also include hallucinations with generative AI and should be included in the first section will all the general information.
- Why all the Tables are in the Supplement? If you use the same variables you could have 2 Tables (Table 1 and Table 2 that regroups Tables S2 to S10) and you could put both in the manuscript.
Minor comments
- 5.2.2 is not “Coronary artery disease” but “traumatic brain injury”
- Reference 13 need the website location.
- Reference 48 has no authors
- Reference 64, it is the first name and not last name that are listed
Author Response
Dear Reviewer
Many thanks for your valuable comments to improve our manuscript “The value of clinical decision support in healthcare: A focus on screening and early detection”. We reply here in a line-by-line fashion to your monita. In addition to this letter, we submitted a version with changes (highlighted in red) and a clean version. We also attached the official "Letter of changes" for you to crosscheck how your and other comments have impacted the structure of the paper. We hope that we could sufficiently address your points. Many thanks for your effort.
The first 3 chapters are generalities that can be regrouped in one. The conclusion of this first part could introduce the next one.
We have regrouped the chapter and created one comprehensive text leading to the conclusion. We now follow a new logic in the paper: 1. General Aspects of CDS plus summary, 2. Algorithm types of CDS through the lens of prevention plus summary. We hope that we achieve a clearer structure with this approach. The summary of chapter 1 remains the old summary of the previous version, while the summary of the second chapter we added at the end of the paper:
“In Lung and CRC, AI models like LungFlag and ColonFlag have demonstrated improved risk prediction, early detection, and cost-effectiveness compared to standard screening methods. Also, AI tools for Fibrosis offer a promising alternative to liver biopsies for assessing liver disease progression. Some of these algorithms have received FDA approval and CE marking, indicating that they meet regulatory standards for clinical use. However, their effectiveness can still be variable, as performance often depends on factors such as the underlying liver disease, patient demographics, and the specific biomarkers incorporated. Novel CAD algorithms show high accuracy and offer a faster cardiac risk detection approach, impacting the clinical workflow in early diagnosis of non-ST-segment elevation myocardial infarction (NSTEMI) in critical care. Their usage will result in faster decision-making on life-saving interventions such as Percutaneous Coronary Intervention and offer superior workflows compared to the standard of care. In TBI, AI models exist that are able to select patients who will benefit from further imaging diagnostics and preventive neurohabilitation while using available resources more efficiently. Algorithms in sepsis can improve time to diagnosis and diagnosis accuracy but the development of such algorithms would greatly benefit from a consensus on the precise definition and management of sepsis. Until such an agreement is reached, it is not possible to make any definitive statements about the performance. Finally, available algorithms in CKD reported superior performance compared to the standard of care, e.g., KDIGO heatmap, in the development and validation cohorts. Despite the potential of AI-based algorithms to innovate healthcare, more research is required to improve generalizability, predictive power and to confirm clinical and economic unity across various populations and healthcare systems. Real-world validation, standardized protocols, and broader testing are required to broaden the use of those tools in secondary prevention. In summary AI-based algorithms play a crucial role in early detection and screening by enhancing risk stratification, especially in oncology. In tertiary prevention, algorithms help manage diseases and prevent complications in conditions such as acute coronary care, traumatic brain injuries, and sepsis.”
In the chapter “Algorithm types of CDSS through the lens of prevention”, a text explaining why this is the focus of the manuscript could be helpful, as well as an explanation on the methodology used to select the CDSS presented and the reason why choosing oncology, acute care and nephrology (include also a small sentence on that in the abstract and summary).
We have included the following paragraphs into the main body of the paper:
“In this section we present contemporary algorithms for the secondary prevention of seven relevant diseases. such as lung cancer, colorectal cancer, hepatocellular cancer, coronary artery disease, traumatic brain injury, sepsis and chronic kidney disease. Those conditions were selected based on global burden of disease because of their significance in Prevalence, Mortality or affecting life quality and health care costs:
Rationale:
Early detection of certain cancers and other chronic diseases through screening can reduce mortality from these conditions by 15 to 20 percent (Ref). Ample evidence exists that secondary prevention can reduce mortality of malignancies by 15 to 20% (Ref). Because lung cancer (18.7%), colorectal cancer (9.3%) and liver cancer (7.8%) remain the leading causes of cancer death globally (Ref) there is an unmet medical need to improve early detection to consecutively benefit survival and reduce health care costs. 2017 US research yielded that early detection can lead to cost saving of $3.47B for CRC and 3.43B for Lung Cancer (Ref). On a global scale coronary artery disease remains the leading cause of death, accounting for approximately 13% of the world’s total deaths (Ref). Similar to the above-mentioned malignancies, the disease greatly benefits from early interventions prior to the occurrence of symptoms. In a global comparison of economic costs, annual expenses of the condition exceeded the total health expenditure per capita by 4.9% to 137.8% (Ref), thus qualifying for early detection and consecutive intervention. Traumatic brain injury (TBI) implies approximately $US 400 billion substantial annual costs to health care systems (Ref) and will remain beyond 2030 the leading cause of disability. Despite implying higher health care burden as compared to neurodegenerative disorders the condition goes almost unrecognized in public health strategies (Ref). Sepsis implies approximately 20% of all global deaths (Ref) of which 50% occur in children between 1 and 4 years (Ref). Because the condition is considered the leading cause of death in critically ill patients with an almost 50% mortality (Ref), the implication on healthcare spent is significant and contributes with 2.65% of the entire healthcare budget (Ref.)] highlighting the necessity for improved prevention. Finally, we included Chronic Kidney Disease (CKD) as a relevant condition into our analysis because approximately 850 million patients suffer globally from the disease, most of them in low-and lower-middle-income countries with no access diagnosis and consecutive treatment (Ref) Bearing in mind that CKD is the third fastest-growing cause of death globally Ref), the condition and epidemiology qualifies well for early detection by innovative algorithms, especially in the light of increasing availability of POC devices.
We included the following sentence into the abstract: “In particular it consolidates research across diseases which imply a major disease and economic burden such as lung cancer, colorectal cancer, hepatocellular cancer, coronary artery disease, traumatic brain injury, sepsis and chronic kidney disease.”
In the Table S2 to S10 that would be helpful to have the same variables presented including “input data” and similar variables as in Table 6 (“Limitations”, “Regulatory Status”, “Clinical Use”).
We have amended all Tables according to the format of Table S6
Chapter 6 (“Challenges and limitations of CDS”) is very short, should also include hallucinations with generative AI and should be included in the first section will all the general information.
Thanks for this important comment. We have included the paragraph below, however we decided to add it to the section Challenges and Limitations, because we see a better fit in this section.
“Generative AI models, which are increasingly being integrated into CDS, are prone to hallucinations—instances where the AI generates incorrect or fabricated information. Contemporary research suggests that hallucinations should be labeled as “confabulations” or better as AI-misinformation, to prevent the stigmatization of AI ®. Addressing AI-based misinformation requires robust verification mechanisms, cross-referencing AI outputs with established medical knowledge, and ensuring that human oversight remains central to decision-making. Also, a model with more parameters that has been trained for longer tends to confabulate less, but this is computationally expensive and involves trade-offs with other Chabot skills, such as an ability to generalize. Training on larger, cleaner data sets helps, but there are limits to what data is available. Developers must prioritize transparency in how generative models operate and train them with high-quality, domain-specific datasets to minimize errors. Furthermore, regulatory frameworks should mandate rigorous testing and validation of generative AI components in CDS systems.”
Why all the Tables are in the Supplement? If you use the same variables you could have 2 Tables (Table 1 and Table 2 that regroups Tables S2 to S10) and you could put both in the manuscript.
We have relocated the tables into the main body of the manuscript. The reason why we initially had them as Supplementary Material was the magnitude of the file and the different format. We have created one table and displayed all different disease areas with the corresponding diseases.
5.2.2 is not “Coronary artery disease” but “traumatic brain injury”
Thank you, we have addressed this mistake
Reference 13 need the website location
Thank you, we have added this.
Reference 48 has no authors
We have added the authors
Reference 64, it is the first name and not last name that are listed
We have cross-checked all References for accuracy.

Reviewer 2 Report
Comments and Suggestions for Authors
Dear Author
The manuscript was not technically sound and not create any impact related to this research.
Author Response
Dear reviewer. We sadly cannot reply to your comment because we do not know more details. However, we attached the Letter of changes based on the other reviewer comments for your reference.

Reviewer 3 Report
Comments and Suggestions for Authors
The manuscript paves attention towards exploring various metrices for diseases in building a early diagnosis system for clinical decision support system.
However, the metrices are not adequate and the deliberated metrices are so generic in nature. No novelty found in terms of proposed metrices. All are well-known and just put forth the percentage.
What is the contribution from authors ? and in what way interpretation may be made for better clarity?
Tables are just displaying the values for comparison.
References are given at the end of each Table. It has to be avoided.
More content, focus towards the contribution of the paper are to be highlighted in Chapter 1 of the manuscript.
More specific Metrices with respect to various novel deep learning architectures are to be studied, and to be presented,.
More content are to be added with novelty.
Author Response
Dear Reviewer
Many thanks for your valuable comments to improve our manuscript “The value of clinical decision support in healthcare: A focus on screening and early detection”. We reply here in a line-by-line fashion to your monita. In addition to this letter, we submitted a version with changes (highlighted in red) and a clean version. We also attached the official "Letter of changes" for you to crosscheck how your and other comments have impacted the structure of the paper. We hope that we could sufficiently address your points. Many thanks for your effort.
Metrices are not adequate and the deliberated metrices are so generic in nature. No novelty found in terms of proposed metrices. All are well-known and just put forth the percentage.
We have updated the description with additional details on quality metrics to ensure high quality CDS. Table S1 was updated to include a new column with examples of specific metrics associated to each algorithm feature. Additionally, the description of each feature was updated to emphasize the need and the utility of these desired features.
The following text was added/changed:
“User acceptance of CDS, regardless if the user is a physician, a nurse, or a patient, is a fundamental requirement for its successful application [R]. Achieving this requires the evaluation of CDS systems against comprehensive quality metrics that not only ensure technical robustness but also promote clinical relevance and ethical integrity. A critical aspect of this evaluation involves identifying the core features of specific clinical challenges and designing performance metrics that enhance the utility of these systems for clinicians (R) This process necessitates close collaboration with healthcare professionals to establish clinically meaningful and practically achievable goals for the algorithms. These goals guide the optimization process by defining what constitutes success in real-world applications. Addressing diverse objectives often involves implementing multiple cost functions and constraints to capture different priorities—such as diagnostic accuracy, timeliness, healthcare costs, and resource capacity. Furthermore, these metrics are vital in identifying and mitigating biases, ensuring equitable performance across varied patient populations. The following table displays seven features that are considered a prerequisite for an ideal algorithm applied in any form of CDS [R]. These features and metrics serve as a foundation for developing, evaluating, and refining CDS systems, ensuring they are reliable, equitable, and seamlessly integrated into clinical workflows.”
What is the contribution from authors ? and in what way interpretation may be made for better clarity?
Thanks for this comment: We have added the following paragraph right at the beginning of the paper to highlight the contribution of the paper. We also describe the two parts of the paper.
“CDS systems have become essential in modern healthcare, increasing patient management through AI and ML. By processing complex, multi-domain data, these systems provide real-time support that improves diagnostic accuracy and personalizes treatment, addressing human limitations in making multi-dimensional care decisions. In the first part of this literature review we provide a comprehensive analysis of CDS systems, emphasizing their role in early detection and prevention. In the second part, we discuss selected diseases with a high disease burden whose prevention, by using AI, may substantially decrease mortality, prevalence, YLDs and healthcare costs. We display various commercially and non-commercially available algorithms as well as their performance parameters and critically discuss their performance with the generally accepted standard of care.”
Tables are just displaying the values for comparison.
The aim of the tables is to highlight available algorithms and their performance parameters that are not accessible in a database but have to be compiled from various academic or commercial websites. To our knowledge there is no comprehensive database available where performance parameters of different algorithms are accessible and comparable. We see value in presenting the reader with the different performance parameters of contemporary available algorithms and putting them into the context of the standard of care. We sadly cannot really respond to this monitum because it was not explicitly mentioned, what should be addressed. We internally assessed the possibility of statistical comparison of algorithm performance as compared to the standard of care but such approach would not be scientifically sound due to the low number of available algorithms. As a result, the only way is a qualitative interpretation, which we have tried to give in the various sections. However, we try to address the reviewers concern by giving the following explanations:
Lung Cancer
- Comparisons between the models and the standard of care (SoC) were detailed in the initial manuscript. The lung cancer section highlights key differences:
- Current guidelines rely on simple metrics like age and smoking history, excluding high-risk groups and leading to high false positives and overdiagnosis.
- Advanced models, such as Lung Flag and PLCOm2012, incorporate broader risk factors for improved stratification, higher accuracy, and earlier detection.
- Risk-based strategies are also more cost-effective, with incremental cost-effectiveness ratios below $100,000 per QALY.
- Table 3 provides performance metrics (e.g., sensitivity, specificity, AUC), demonstrating the models' superiority over the standard of care.
- Definition of specificity and sensitivity:
- Definitions for specificity and sensitivity have been added to the manuscript as follows:
“Sensitivity and specificity are critical metrics of screening effectiveness; while LDCT excels in identifying true positive cases of lung cancer, its low specificity results in high false positives and overdiagnosis rates, leading to unnecessary tests and invasive procedures [43].”
Colorectal Cancer
- Definition of specificity and sensitivity:
- Definitions for specificity and sensitivity have been added to the manuscript as follows:
“Most models in Table 3 demonstrate moderate sensitivity (the ability to correctly identify individuals with CRC or advanced neoplasia) and specificity (the ability to correctly identify those without CRC), indicating room for improvement. For example, APCS has a sensitivity of 42% and specificity of 86%, which is sufficient to reduce unnecessary colonoscopies but may miss a significant portion of high-risk individuals.”
- Model comparisons:
- Comparisons between the models and standard-of-care (SoC), as well as among the models themselves, were already described in the initial manuscript version.
- Hybrid approach emphasis:
- The last paragraph of the Colorectal Cancer section has been amended to highlight the value of a hybrid approach that combines risk models with standard-of-care methods.
Hepatocellular Carcinoma
- A description has been added before former Table 5:
” Several algorithms that combine various biomarkers with clinical features have been developed and tested. However, most of these (with one exception) remain research tools, as they have not received regulatory approval due to the heterogeneous results from clinical performance studies, which have been conducted in different populations and with varying cutoff values.
- Additional background information has been added for former Table 6 (now Table 3)
Traumatic Brain Injury
- Definition for specificity and sensitivity:
- Definitions for specificity and sensitivity have been added to the manuscript as follows:
“(...) Nevertheless, not without limitations. Despite this commercially available and validated tool (Table S8, row 1) shows high sensitivity equal to 95,8% (blood-based biomarkers present concretely high ability and accuracy to correctly identify subjects with peri-axonal lesions after mTBI), the relatively low specificity not exceeding 40,4% addresses the potential need to perform the best neuroimaging such as MRI to confirm brain parenchymal changes, correctly suggested by increased levels of blood-based biomarkers and not detected by CT scan. (...)”
- Model comparisons:
- The most promising diagnostic and risk-stratification tools are available as research based large patients’ data sets but not yet commercially available and validated. At this point in time, a concrete model comparison is not yet possible even though the Table 3 is showing all the efforts in the last few years focused on supporting physicians in the ED by correctly identifying patients with brain parenchymal changes after mTBI (secondary prevention) and by consequently preventing neurological disabilities (tertiary prevention).
- Added emphasis:
- The last paragraphs have been implemented to add emphasis on all the efforts, shown in the Table 3, made in the last years and still ongoing to impact the healthcare systems and the societies.
CKD
- We added to the main body the following sentences:
- “Overall, these novel algorithms have moderate to high accuracy (AUC above 0.78), and have great potential to identify patients at high risk at an early stage for early intervention. However, there is limited external validation in other markets and populations, as well as limited evidence on their effectiveness in real-world clinical practice. Further research is warranted to demonstrate the real-world effectiveness and generalizability of these novel algorithms.”
CAD
- Added to the main body:
- “These novel CAD algorithms demonstrate high accuracy (AUC 0.87 and 0.95–0.98) and strong potential for early cardiac risk identification and fostering faster life-saving interventions. However, adoption of CDS often faces practical inertia, and their clinical impact on workflow efficiency and outcomes requires more real-world data across different healthcare systems and populations to confirm their positive impact and generalizability.”
Sepsis
- Added to the main body the discussion about the latest trend in the field (moving from Early Warning System to truly regulated SaMDs) and what this means in terms of regulatory evidence. Further model comparisons cannot be done because many study parameters are not equivalent, e.g. in terms of study population and sepsis definition. Additionally, an emphasis has been put on the importance of the clinical validation of these models in the true clinical setting.
References are given at the end of each Table. It has to be avoided.
We embedded the Tables in the main manuscript and merged disease-area specific tables (Tables 3-10) to one big table (now Table 3).
More content, focus towards the contribution of the paper are to be highlighted in Chapter 1 of the manuscript.
This comment is similar to comment 7 and 8 (Letter of changes attached). To address both reviewer comments at the same time we have merged Chapter 1, 2, and 3, conclude the first chapter with parts of the summary and added the following paragraph at the very beginning of the paper:
“CDS systems have become essential in modern healthcare, increasing patient management through AI and ML. By processing complex, multi-domain data, these systems provide real-time support that improves diagnostic accuracy and personalizes treatment, addressing human limitations in making multi-dimensional care decisions. In the first part of this literature review we provide a comprehensive analysis of CDS systems, emphasizing their role in early detection and prevention. In the second part, we discuss selected diseases with a high disease burden whose prevention, by using AI, may substantially decrease mortality, prevalence, YLDs and healthcare costs. We display various commercially and non-commercially available algorithms as well as their performance parameters and critically discuss their performance with the generally accepted standard of care.”
Furthermore we added the following sentence to the abstract for a better understanding:
“In particular it consolidates research across diseases which imply a major disease and economic burden such as lung cancer, colorectal cancer, hepatocellular cancer, coronary artery disease, traumatic brain injury, sepsis and chronic kidney disease.”
An even more detailed explanation about the chosen conditions we added in part 2 of the paper (please see how we address reviewer comment 8).
More specific Metrices with respect to various novel deep learning architectures are to be studied, and to be presented.
We have updated the description with additional details on quality metrics to ensure high quality CDS. Table S1 was updated to include a new column with examples of specific metrics associated to each algorithm feature. Additionally, the description of each feature was updated to emphasize the need and the utility of these desired features.
The following text was added/changed:
“User acceptance of CDS, regardless if the user is a physician, a nurse, or a patient, is a fundamental requirement for its successful application [R]. Achieving this requires the evaluation of CDS systems against comprehensive quality metrics that not only ensure technical robustness but also promote clinical relevance and ethical integrity. A critical aspect of this evaluation involves identifying the core features of specific clinical challenges and designing performance metrics that enhance the utility of these systems for clinicians (R) This process necessitates close collaboration with healthcare professionals to establish clinically meaningful and practically achievable goals for the algorithms. These goals guide the optimization process by defining what constitutes success in real-world applications. Addressing diverse objectives often involves implementing multiple cost functions and constraints to capture different priorities—such as diagnostic accuracy, timeliness, healthcare costs, and resource capacity. Furthermore, these metrics are vital in identifying and mitigating biases, ensuring equitable performance across varied patient populations. The following table displays seven features that are considered a prerequisite for an ideal algorithm applied in any form of CDS [R]. These features and metrics serve as a foundation for developing, evaluating, and refining CDS systems, ensuring they are reliable, equitable, and seamlessly integrated into clinical workflows.”
More content are to be added with novelty.
We added more contemporary research in the section “coronary artery disease”. We added additional contemporary and authoritative research in the section “Sepsis” demonstrating the benefit of AI.

Round 2
Reviewer 1 Report
Comments and Suggestions for Authors
Congratulations to the authors. They did an extensive revision and addressed all my comments. Only one small minor change: page 6 line 443, remove "g^" at liver g^fibrosis".
Author Response
Dear Reviewer.
Many thanks for your comment. We now have done a final check on grammar and language with a native speaker. For your reference, we attach the final clean version of the paper. In the main submission mask we also submit a file with track changes.
Once again, we kindly thank you for your help and support during the process.

Reviewer 2 Report
Comments and Suggestions for Authors
Dear Authors
Now paper was fine. Authors incorporate all changes
Author Response

(The authors gave the same response as above.)

Reviewer 3 Report
Comments and Suggestions for Authors
All the comments given in the earlier version has been addressed by the Authors. However, authors are advised to proof-read the language style in the formation of sentences. Though it not appear at many places. Proof read the manuscript would be ideal.
Comments on the Quality of English LanguageAll the comments given in the earlier version has been addressed by the Authors. However, authors are advised to proof-read the language style in the formation of sentences. Though it not appear at many places. Proof read the manuscript would be ideal.
Author Response

(The authors gave the same response as above.)
